# Efficacy and Safety of the Addition of Internal Mammary Irradiation to Standard Adjuvant Radiation in Early-Stage Breast Cancer: A Systematic Review and Meta-Analysis

Yasmin Korzets [1,2], Dina Levitas [3,4], Ahuva Grubstein [2,5], Benjamin W. Corn [3,4], Eitan Amir [6] and Hadar Goldvaser [3,4,*]

1   Institute of Oncology, Tel Aviv Sourasky Medical Center, Weizmann St. 6, Tel-Aviv 6423906, Israel
2   Sackler Faculty of Medicine, Tel Aviv University, Chaim Levanon St. 30, Tel-Aviv 6997801, Israel
3   Oncology Institute, Shaare Zedek Medical Center, Shmuel Bait St. 12, Jerusalem 9103102, Israel
4   Faculty of Medicine, Hebrew University, Ein Kerem. P.O. Box 12271, Jerusalem 9112102, Israel
5   Department of Imaging, Rabin Medical Center, Jabotinsky St. 39, Petah Tikva 4941492, Israel
6   Princess Margaret Cancer Centre, University of Toronto, 610 University Ave, Toronto, ON M5G 2C1, Canada
*   Correspondence: hadargo@szmc.org.il; Tel.: +972-2-6555361

**Abstract:** Background: Existing data on adding internal mammary nodal irradiation (IMNI) to the regional nodal fields are inconsistent. Methods: Randomized trials investigating the addition of IMNI to standard adjuvant radiation were identified. Hazard ratios (HRs) and 95% confidence intervals (CI) were extracted for overall-survival (OS), breast cancer specific-survival (BCSS), and disease-free survival (DFS) as well as distant-metastasis free survival (DMFS). The odds ratios (ORs) for regional and loco-regional recurrence, non-breast cancer mortality, secondary non-breast cancer, contralateral breast cancer, and cardiovascular morbidity and mortality were also extracted. Results: Analysis included five trials comprising 10,994 patients, predominantly with higher risk, lymph node positive disease. Compared to the control group, IMNI was associated with significant improvement in OS (HR = 0.91, $p = 0.004$), BCSS (HR = 0.84, $p < 0.001$), DFS (HR = 0.89, $p = 0.01$), and DMFS (HR = 0.89, $p = 0.02$). IMNI was also associated with reduced odds for regional (OR = 0.58, $p < 0.001$) and loco-regional recurrence (OR = 0.85, $p = 0.04$). The odds for cardiotoxicity were not statistically significantly higher (OR = 1.23, $p = 0.07$). There were comparable odds for cardiovascular mortality (OR = 1.00, $p = 1.00$), non-breast cancer mortality (OR = 1.05, $p = 0.74$), secondary cancer (OR = 0.95, $p = 0.51$), and contra-lateral breast cancer (OR = 1.07, 95% 0.77–1.51, $p = 0.68$). Conclusions: Compared to the control group, the addition of IMNI in high-risk patients is associated with a statistically significant improvement in survival, albeit with a magnitude of questionable clinical meaningfulness.

**Keywords:** breast cancer; radiotherapy; internal mammary irradiation; regional nodal irradiation

## 1. Introduction

The role of adjuvant radiation in improving outcomes for early-stage breast cancer patients is well-established [1,2]. Whole-breast radiotherapy following breast-conserving surgery significantly improves locoregional control and overall survival (OS) [1]. Postmastectomy adjuvant radiotherapy is also associated with improved outcome, mainly in node positive disease [2].

Breast cancer has the potential to spread to regional lymph nodes, most notably the axillary, peri-clavicular, and internal mammary lymph nodes (IMN). Historic radical mastectomy series showed substantial incidence of IMN involvement, ranging between 5 and 10% in axillary node-negative patients and 25–40% in axillary node-positive individual [3,4]. Rates of IMN involvement are also greater in patients with larger tumors, high burden of axillary involvement, and primary breast tumors situated centrally or medially [5,6].

Current standard of care supports regional nodal irradiation (RNI) in most node positive breast cancer patients, in addition to whole breast or chest wall irradiation [7–9]. RNI comprises different fields of radiation including the axilla, supra/infraclavicular lymph nodes, and internal mammary nodes. Typically, decisions on whether to include all or only part of these fields is based on the disease staging at presentation, tumor subtype and location, the extent of axillary and breast surgery, patient age and comorbidities, and response to neoadjuvant systemic therapy when applicable. The addition of IMN irradiation (IMNI) remains controversial. While several pivotal trials have shown that adding IMNI improves locoregional control and disease-free survival (DFS) [10,11], improvements in OS have been inconsistent in trials [12–15]. Concern has also been raised that higher doses to the heart and lungs with the addition of IMNI may contribute to toxicity and increase non-breast cancer death [16].

It remains unclear whether the risk and benefit balance of IMNI justifies its routine use in all high-risk breast cancer patients receiving adjuvant radiation. The objective of this study was to conduct a meta-analysis to explore the efficacy and safety of adding IMNI to adjuvant radiation therapy in early-stage, high risk breast cancer. We also aimed to identify whether a specific subgroup may benefit more from IMNI.

## 2. Materials and Methods

### 2.1. Literature Review and Study Identification

The analysis was conducted following the Preferred Reporting Items for Systematic Reviews and Meta-Analyses (PRISMA) guidelines. This review was not registered. A librarian-directed, comprehensive literature search was conducted to identify all RCTs published between January 2005 and May 2022 that evaluated the addition of IMNI to standard adjuvant radiation in early-stage breast cancer. The search was conducted in MEDLINE (Host: PubMed) using the query: ('breast cancer') AND (radiation OR radiotherapy) AND ('lymph node'). The following filters were applied: clinical trial, meta-analysis, randomized controlled trial, systemic review. A further similar search was conducted in EMBASE using the query: 'breast cancer' AND (radiation OR radiotherapy) AND 'lymph node' AND ((systematic review)/lim OR (meta analysis)/lim OR (controlled clinical trial)/lim OR (randomized controlled trial)/lim) AND (2005–2022)/py AND (english)/lim.

The results of both searches were combined and the duplicates were removed. Eligibility criteria were prospective studies investigating the addition of IMNI to standard irradiation in early-stage breast cancer. Studies were included if data on the efficacy or toxicity were reported. Both randomized phase 2 and phase 3 studies were included. One additional study [14], a prospective non-randomized study that relied on 'natural random allocation' to IMNI, depending on tumor side (right sided tumors for IMNI and left sided tumors without IMNI), was also included. The search was restricted to the English language.

### 2.2. Data Extraction

Data were extracted from primary publications and their appendices by two independent reviewers (YK and DL). Discrepancies between both reviewers were discussed and resolved together with a third reviewer (HG). For studies with several publications, the most updated results were extracted. Extracted data included the year of publication, median duration of follow-up, number of patients, and their available demographic information. Details on the treatment given in each arm were also collected including data on the field and dose of radiation and on whether a boost was delivered. Additionally, data on of the systemic treatment and on the type and extent of surgery were extracted. Trial-level summary data were collected for tumor characteristics including the tumor side (right or left), tumor location (lateral or central/medial), tumor size, extent of nodal involvement and presence of extracapsular extension, hormone receptor and human epidermal growth factor receptor 2 (HER2) expression, histological subtype (invasive ductal carcinoma compared to other), and tumor grade.

Hazard ratios (HRs) and respective 95% confidence intervals (CI) were extracted for the following pre-specified efficacy endpoints: OS, DFS, breast cancer specific survival (BCSS), and distant metastases free survival (DMFS). If OS was not reported explicitly, the HR was estimated from Kaplan–Meier curves (if reported) using the Parmar Toolkit method [17]. OS results were also extracted for prespecified subgroups defined by age, menopausal status, tumor side, tumor location, receptor expression (i.e., hormone receptor positive, HER2 positive, triple negative disease), tumor size, extent of nodal involvement, grade, and number of lymph nodes removed.

For toxicity, the following pre-specified data were collected: pneumonitis, cardiovascular morbidity and mortality, and secondary non-breast cancer malignancy. Data on locoregional recurrence, regional recurrence, non-breast cancer mortality, and contralateral breast cancer were extracted when reported.

*2.3. Data Synthesis and Statistical Analysis*

The primary analysis was the association between IMNI and OS. For all endpoints, the HR and associated 95% CI were pooled in a meta-analysis using RevMan 5.4 (The Cochrane Collaboration, Copenhagen, Denmark). Subgroup analyses were conducted in order to evaluate the efficacy of IMNI on OS in several subgroups defined by patients and tumor characteristics. Differences between defined subgroups were assessed using methods described by Deeks et al. [18]. Meta-regression was performed using SPSS version 25 (IBM Corp, Armonk, NY, USA) for continuous variables, using the weighted least squares (mixed effect) function. As the number of included studies was small, meta-regression analyses were interpreted quantitatively using the Burnand criteria [19] rather than to infer associations based on statistical significance [20,21].

Subgroup analyses explored differences between the included studies with respect to the patient and tumor characteristics as well as the study characteristics such as the inclusion criteria (inclusion of node positive only compared to both node negative and node positive disease) and fields of radiation to the control group (breast/chest wall only compared in addition to other regional lymph node fields except for IMN).

For the toxicity endpoints, and for regional, locoregional recurrence, and contralateral breast cancer, the odds ratio (OR) and associated 95% CI were computed and subsequently pooled in a meta-analysis. When the absolute event rates in the experimental and control groups were smaller than 1% in at least one study, the Peto one-step OR [22] was used to pool estimates of OR, otherwise the Mantel–Haenszel OR method was used [23]. Cochran $Q$ and $I^2$ statistics was used to report the statistical heterogeneity. Statistically significant heterogeneity was defined as a Cochran $Q$ $p < 0.10$ or $I^2$ greater than 50%. Fixed-effect modeling was performed in analyses without statistically significant heterogeneity, otherwise random-effects modeling was used. The following sensitivity analyses were conducted to address heterogeneity excluding studies utilizing natural randomization, studies in which all patients had breast conserving surgery (rather than mastectomy), and any study in which the outcomes were estimated rather than reported explicitly. Finally, due to clinical heterogeneity between studies, any studies pooled initially using fixed effects were repeated using random-effects modeling.

Statistical significance was defined as $p < 0.05$. No corrections were made for multiple significance testing.

## 3. Results

The search identified 1372 studies initially. An additional study was identified by Google [13]. After the removal of duplicates and studies not meeting the inclusion criteria, seven publications reporting on five independent studies were included (see Figure 1 for the study selection schema) [10–15,24]. An additional study reported on updated toxicity, but these data were not relevant to the analysis [25]. The included studies comprised 10,994 patients. Individual study characteristics and details on the radiation treatment in each arm are shown in Table 1. In three studies, the control group was treated with

radiation to the regional lymph nodes other than the IMN [13,14,24] and in two studies, the control group was treated with radiation to the breast or chest wall only [10,11]. About half of the included women had left sided tumor. Two studies included only node positive disease [14,24] and in three studies, node negative disease was also allowed [10,11,13]. Either mastectomy or breast conserving surgery were allowed in three studies [10,13,24], one study included only women who had a mastectomy [13], and one study included only women who had a lumpectomy [11]. Other details on the local and systemic treatment are reported in Table 2.

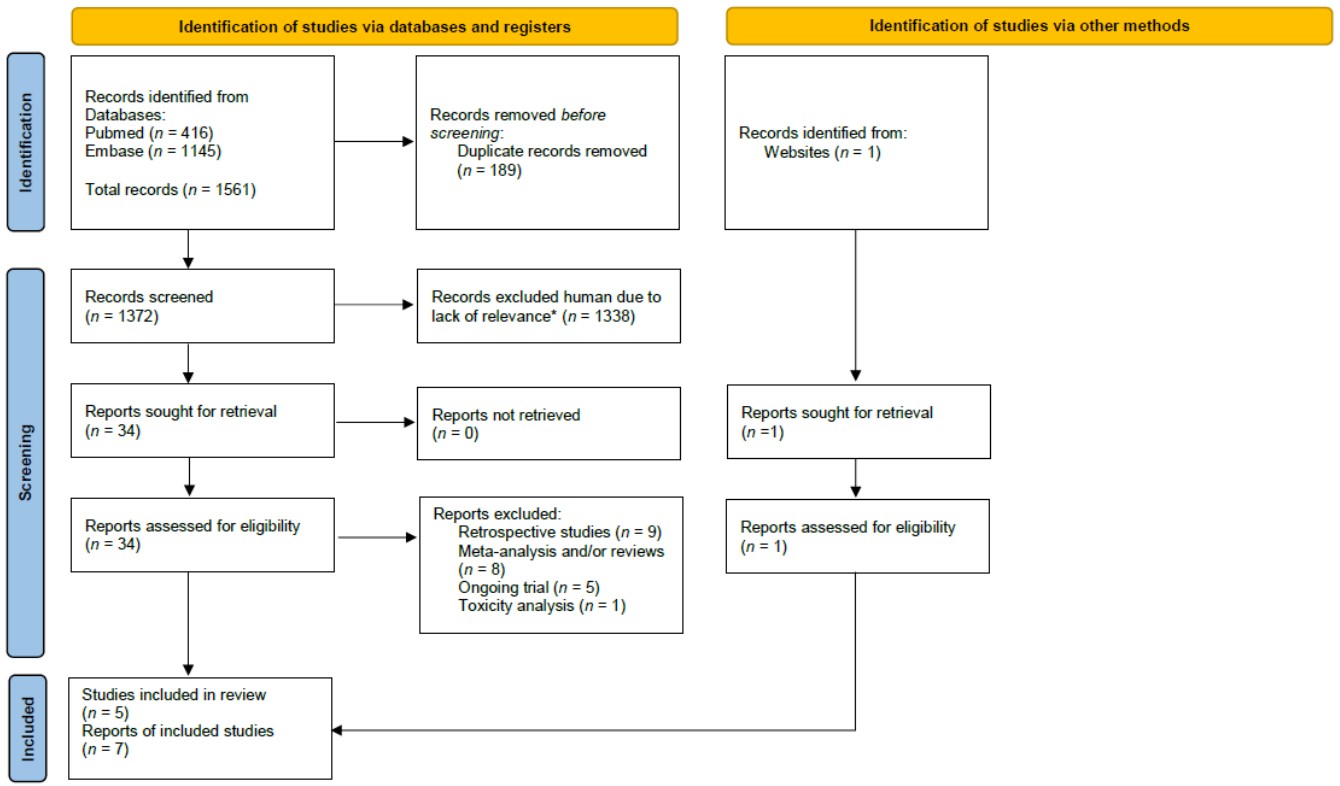

**Figure 1.** The study selection schema.

**Table 1.** The characteristics of the included studies.

| Trial/Median Follow-Up | Treatment Arms | Sample Size (*n*) | Age (yr), Median (Range) | Left Sided Tumor, % | Medial/ Central Tumor, % | Tumor Size [1] | Nodal Status [1] | Receptor Status | Histology (Grade and Histological Subtype) |
|---|---|---|---|---|---|---|---|---|---|
| Kim et al., 2021 [24], 100.4 months | Experimental arm: Radiotherapy to breast/CW+ supraclavicular fossa+ IMNI. 1.8–2 Gy/fraction, total dose of 45–50.4 Gy. Sequential tumor bed boost was allowed. Control: Radiotherapy to breast/CW+ supraclavicular fossa. 1.8–2 Gy/fraction, total dose of 45–50.4 Gy. Sequential tumor bed boost was allowed. | 735 | 48 (28–77) | 50% | 42% | T1 31%, T2 56%, T3 12%, T4 1% | N1 41% N2 37% N3 22% | ER+: 71% HER2+: 23% | Grade 3: 46% IDC: 92% |
| Thorsen 2016, 2022 [14,15], 177.6 months | Experimental (Right-sided breast cancer): IMN treated with an anterior electron field or by inclusion in tangential photon fields. Dose to breast/chest wall, scar, supraclavicular nodes, infraclavicular nodes, and axillary levels II to III was 48 Gy/24 fractions. Control (Left-sided breast cancer): Dose to breast/chest wall, scar, supraclavicular nodes, infraclavicular nodes, and axillary levels II to III was 48 Gy/24 fractions. | 3089 | 56 (22–70) | 52% | 40% | T1 41.5%, T2 52%, T3 6.5%, unknown <1% | N1 59% N2 26% N 15% | ER+: 80% HER2+: NA | Grade 3: 28% IDC: 86% |
| Whelan 2015 [11], 114 months | Experimental: Breast and the ipsilateral IMN (upper three intercostal spaces) + supraclavicular fossa and axillary lymph nodes. 50 Gy/25 fractions. Control: Whole-breast irradiation alone 50 Gy/25 fractions. | 1832 | 53 (26–84) | NR | 38% | T1 52%, T2 47%, T3 1% | N0 10% N1 85% >N2 5% | ER+: 75% HER2+: NA | Grade 3: 43% IDC: NA |

**Table 1.** *Cont.*

| Trial/Median Follow-Up | Treatment Arms | Sample Size (*n*) | Age (yr), Median (Range) | Left Sided Tumor, % | Medial/ Central Tumor, % | Tumor Size [1] | Nodal Status [1] | Receptor Status | Histology (Grade and Histological Subtype) |
|---|---|---|---|---|---|---|---|---|---|
| Poortmans 2015, 2020 [10,12], 188.4 month | Experimental: Breast/CW and internal mammary-medial Supra lymph nodes. 50 Gy/25 fractions. Control: Breast/CW only. 50 Gy/25 fractions. | 4004 | 54 (22–75) | 49% | 66% | T1 60% T2 36% T3 4% | N0 44% N1 43% N2 10% N3 3% | ER+: 74% HER2+ NA | Grade 3: NA IDC: NA |
| Hennequin 2013 [13], 135.6 month | Experimental: CW and supraclavicular fossa + IMN (first 5 intercostal spaces) Control: CW and supraclavicular fossa. | 1334 | NR | 53% | 65% | T1 35% T2 55% T3 9% | N0 25% N1 44% N2–3 31% | ER+: 88% HER2+: NA | Grade 3: 31% IDC: NA |

[1] Tumor size and nodal status: As defined by the American Joint Committee on Cancer (AJCC) 7th edition abbreviations: ALND—axillary lymph node dissection; IDC—invasive duct carcinoma; NA—not available; WBI—whole breast irradiation; CW—chest wall.

**Table 2.** Data on the local and systemic from included studies.

| Trial | Surgery Type | Number of Lymph Nodes Removed, Median (Range) | Boost After Lumpectomy (%) | Chemotherapy (%) | Anti HER2 Therapy (%) | Endocrine Therapy (%) |
|---|---|---|---|---|---|---|
| Kim et al., 2021 [24] | Mastectomy: 49.9% BSC: 50.1% | 17 (4–53) | 97.5% | 98.9% | 23.8% HER2 positive 94.1% received anti her2 therapy | 94.2% |
| Thorsen 2016, 2022 [14,15] | Mastectomy: 65.3% BSC: 34.7% | 16 (13–22) | 12.4% | 18.9% | NA | 100% |
| Whelan 2015 [11] | Mastectomy: 0% BSC: 100% | 12 (8–16), 1–9 32.6%, >10 67.4% | 33.3% | 9% | NA | 98.4% |
| Poortmans 2015, 2020 [10,12] | Mastectomy: 23.9% BSC: 76.1% | <10 23.6%, ≥10 76.3% | NA | 54.8% | NA | 81% |
| Hennequin 2013 [13] | Mastectomy: 100% BSC: 0% | <10 37.4%, ≥10 62.6% | NA | 61% | NA | 100% |

Results of the pre-specified efficacy and toxicity and the studies that were included in each analysis are shown in Table 3. Data on OS were reported explicitly in four studies [11,12,15,24] and were calculated in one study [13]. Compared to the control group, the addition of IMNI was associated with significantly improved OS (HR = 0.91, 95% CI 0.85–0.97, *p* = 0.004), see Figure 2A. Among the studies reporting the absolute number of deaths [11,12,15,24], pooling all such events demonstrated that the absolute reduction was −1.9%, translating to a number needed to treat 52. IMNI was also associated with significant improvement in BCSS (HR = 0.84, 95% CI 0.77–0.92, *p* < 0.001), DFS (HR = 0.89, 95% CI = 0.82–0.98, *p*= 0.01), and DMFS (HR = 0.89, 95% 0.81–0.98, *p* = 0.02), see Figure 2B–D. Compared to the control group, IMNI was associated with lower odds for loco-regional recurrence (OR = 0.85, 95% CI 0.72–1.00, *p* = 0.04, absolute difference −0.82%, number needed to treat 122) and regional recurrence (OR = 0.58, 95% CI 0.44–0.75, *p* < 0.001, absolute difference −1.26%, number needed to treat 79), see Figure 3.

**Table 3.** A summary of the efficacy and toxicity results and sensitivity.

| | Primary Analysis, HR, 95% CI | Analysis with Random Effect | Excluding the DBCG Study [14,15] | Excluding Study that All Patients Had BCS [11] | Excluding the Study all Patients Had Mastectomy [13] | Included Studies |
|---|---|---|---|---|---|---|
| DFS | 0.89 (0.82–0.98), *p*= 0.01 | 0.87 (0.76–0.99) | NA | 0.92 (0.84–1.02) | NA | [11,12,24] |
| OS | 0.91 (0.85–0.97), *p* = 0.004 | 0.91 (0.85–0.97) | 0.94 (0.86–1.02) | 0.91 (0.84–0.97) | 0.90 (0.84–0.97) | [11–13,15,24] |
| BCSS | 0.84 (0.77–0.92), *p* < 0.001 | 0.84 (0.77–0.92) | 0.80 (0.71–0.91) | 0.85 (0.77–0.93) | NA | [11,12,15,24] |
| DMFS | 0.89 (0.81–0.98), *p* = 0.02 | 0.87 (0.77–0.99) | NA | 0.92 (0.82–1.02) | NA | [11,12,24] |
| | **Primary analysis, OR, 95% CI** | | | | | |
| Loco-regional recurrence | 0.85 (0.72–1.00), *p* = 0.04 | 0.85 (0.68–1.05) | 0.81 (0.68–0.96) | 0.90 (0.75–1.07) | NA | [11,12,15,24] |
| Regional recurrence | 0.58 (0.44–0.75), *p* < 0.001 | 0.54 (0.34–0.85) | 0.58 (0.43–0.76) | 0.64 (0.49–0.85) | NA | [11,12,15,24] |
| Secondary cancer | 0.95 (0.82–1.10), *p* = 0.51 | 0.95 (0.82–1.10) | 1.01 (0.85–1.21) | 0.91 (0.77–1.07) | NA | [11,12,15] |
| Cardiotoxicity | 1.23 (0.99–1.53), *p* = 0.07 | 1.23 (0.99–1.53) | NA | NA | 1.22 (0.97–1.54) | [12,13,24] |
| Cardiovascular mortality | 1.00 (0.69–1.46), *p* = 1.00 | 1.00 (0.69–1.46) | 0.98 (0.63–1.51) | 1.00 (0.66–1.52) | NA | [11,12,15,24] |
| Non-breast cancer related mortality | 1.05 (0.79–1.41), *p* = 0.74 | NA | 1.18 (0.98–1.44) | 1.04 (0.70–1.57) | NA | [11,12,15,24] |
| Contralateral breast cancer | 1.07 (0.77–1.51), *p* = 0.68 | NA | 0.90 (0.71–1.14) | 1.11 (0.68–1.79) | NA | [11,12,15] |

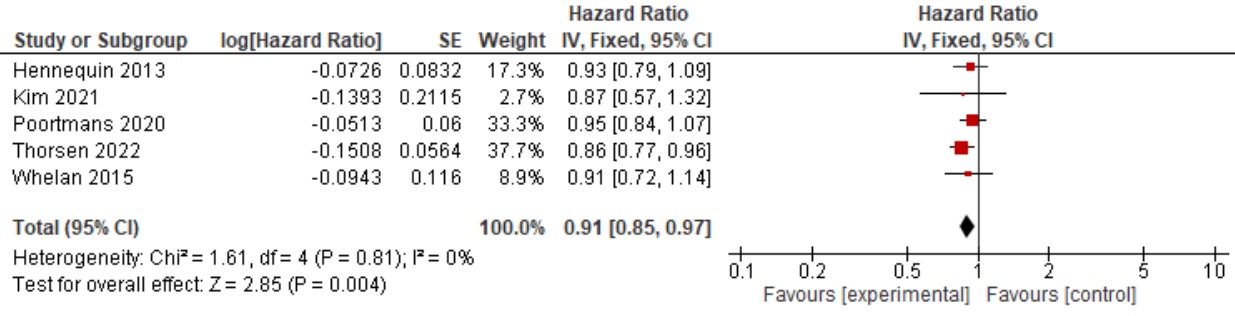

**(A)**

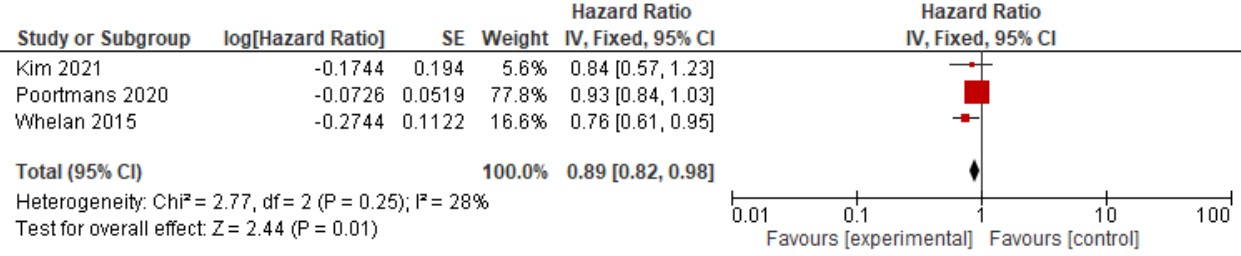

**(B)**

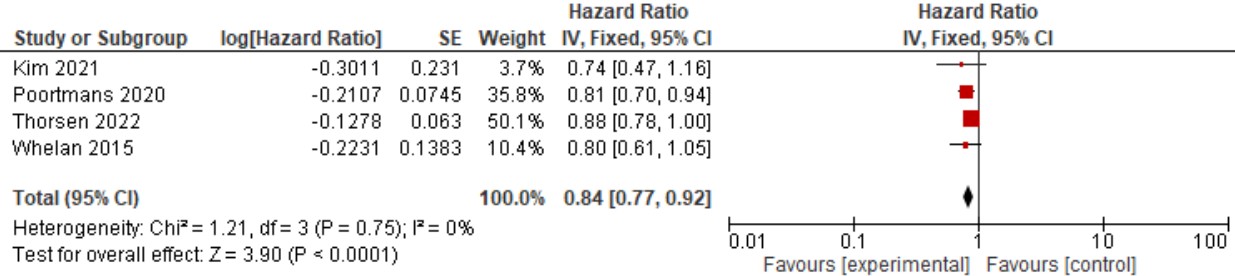

**(C)**

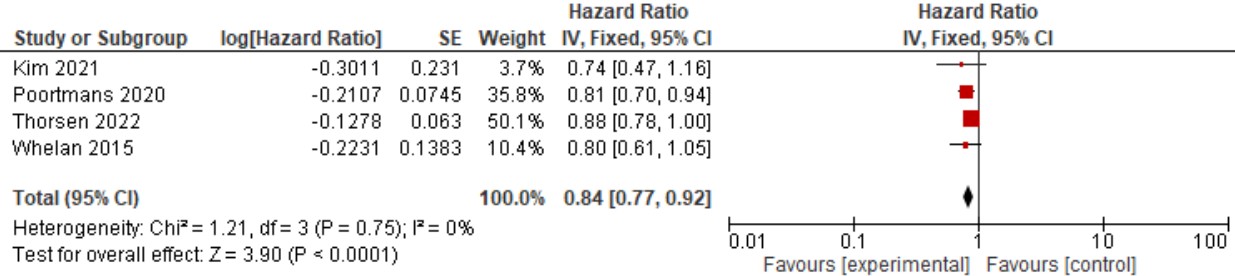

**(D)**

**Figure 2.** Forest plots for the efficacy, hazard ratio for: (**A**) OS; (**B**) DFS; (**C**) BCSS; (**D**) DMFS. The weight of each trial in the meta-analysis is represented by the size of the squares. The estimated pooled effect is represented by a diamond shape. *p* values are two-sided. References: Hennequin 2013 [13], Kim et al., 2021 [24], Poortmans 2020 [12], Thorsen 2022 [15], Whelan 2015 [11].

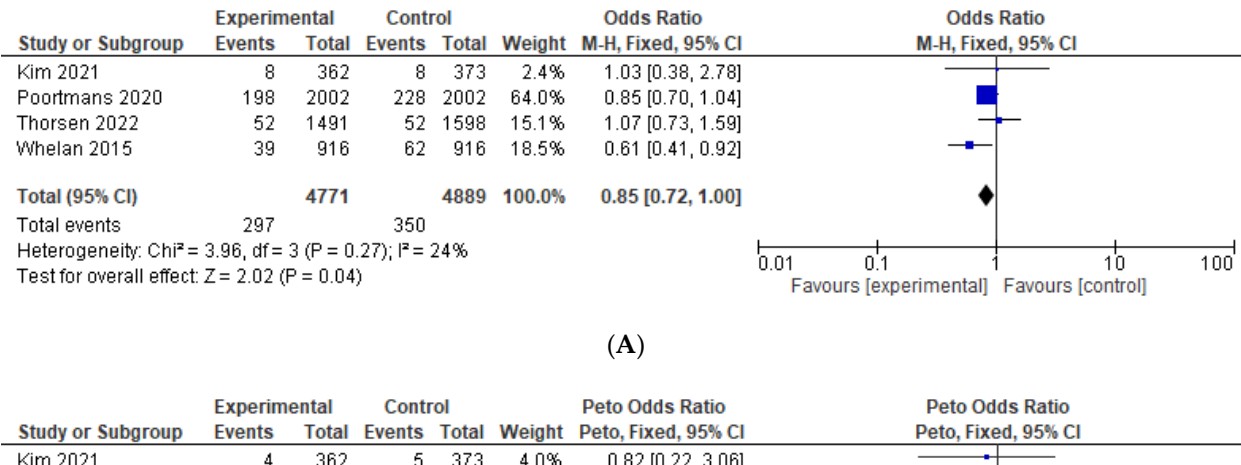

**(A)**

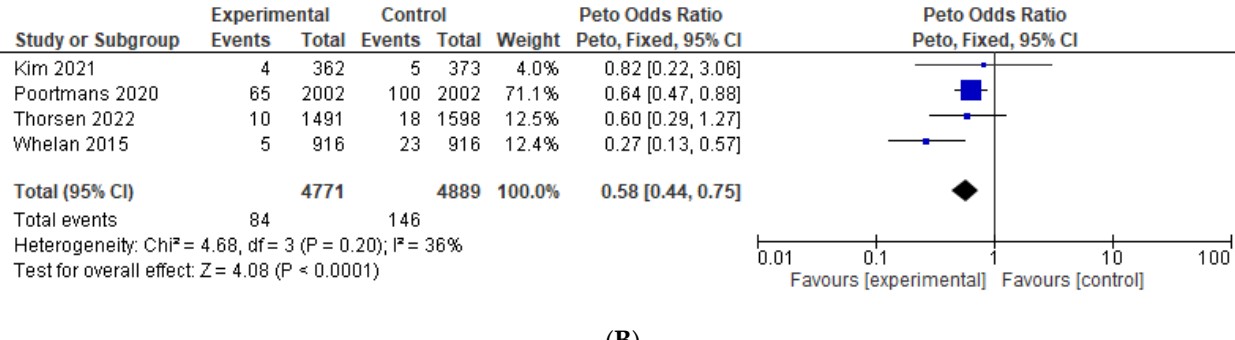

**(B)**

**Figure 3.** Forest plots for the efficacy, odds ratio for: (**A**) loco-regional recurrence; (**B**) regional recurrence. The weight of each trial in the meta-analysis is represented by the size of the squares. The estimated pooled effect is represented by a diamond shape. *p* values are two-sided. References: Hennequin 2013 [13], Kim et al., 2021 [24], Poortmans 2020 [12], Thorsen 2022 [15], Whelan 2015 [11].

OS results for medial/central tumors were available in four studies [11,12,15,24] and for lateral tumors in three studies [11,12,15]. Tumor location had no impact on IMMI benefit, and IMNI was associated with an improvement in OS in both medial/central tumors (HR = 0.89, CI 0.80–0.99, *p* = 0.03) and lateral tumors (HR = 0.90, CI 0.81–1.00, *p* = 0.06, *p* = 0.05), see Figure 4A. Results of OS by the extent of nodal disease are present in Figure 4B. Overall, IMNI had a similar relative benefit in both node negative, 1–3 or more extensive nodal involvement. Data on OS by the number of lymph nodes removed (<10 compared to ≥10 lymph nodes) were available only in two studies [11,12]. The magnitude of benefit from IMNI was higher in women with a low number of axillary lymph nodes removed compared to women with ≥10 lymph nodes removed (HR = 0.83, 95% CI 0.56–1.22, and HR = 0.95, 95% CI 0.80–1.12, respectively). The difference between these subgroups was not statistically significant (*p* for the subgroup difference 0.53). Data on OS by age, menopausal status, tumor size, receptor expression, and grade were limited as such were not pooled into the meta-analysis.

The magnitude of the relative benefit of IMNI on OS was higher in studies including only node positive disease (HR = 0.86, CI 0.77–0.96) compared to studies also including node negative disease (HR = 0.94, CI 0.86–1.02), but this difference was not significant (*p* for subgroup difference 0.22), see Figure S1A. The magnitude of benefit on OS was also comparable in studies where the radiation to the control group comprised the breast or chest wall only and studies where the control group was treated with radiation to other regional lymph nodes (HR = 0.94, 95% CI 0.86–1.05 and HR = 0.87, 95% CI 0.79–0.96, respectively, *p* for the subgroup difference 0.24), see Figure S1B.

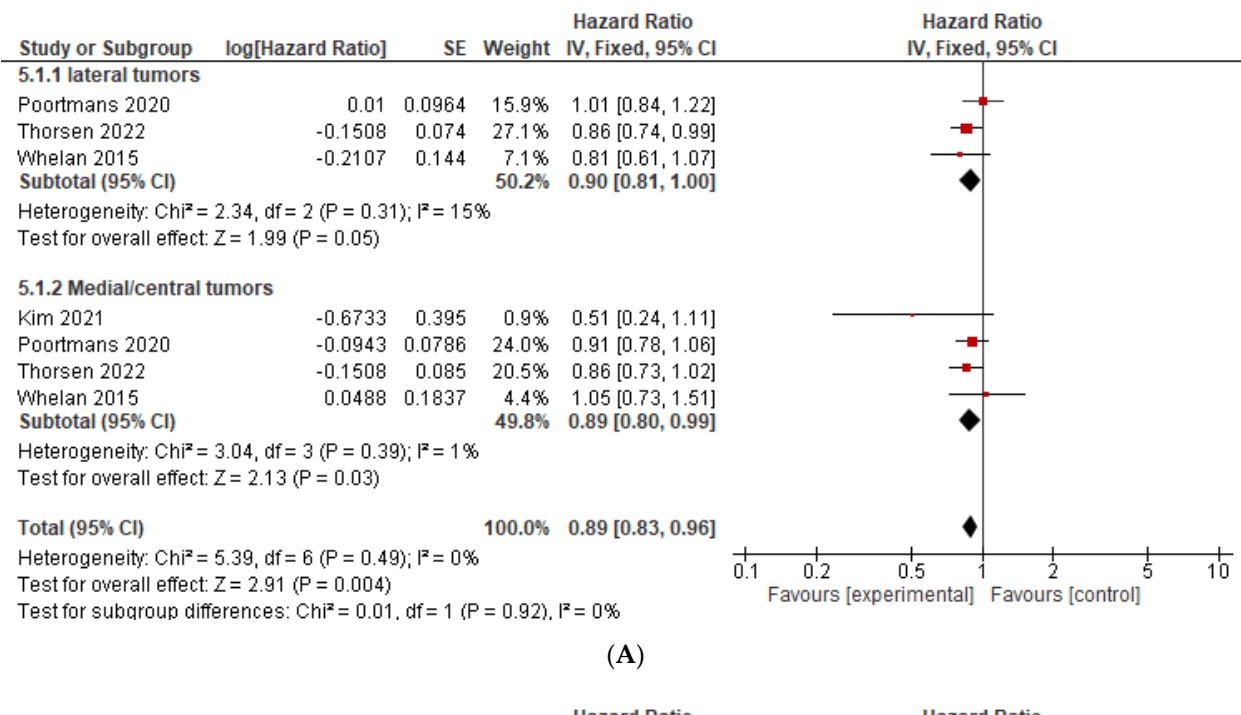

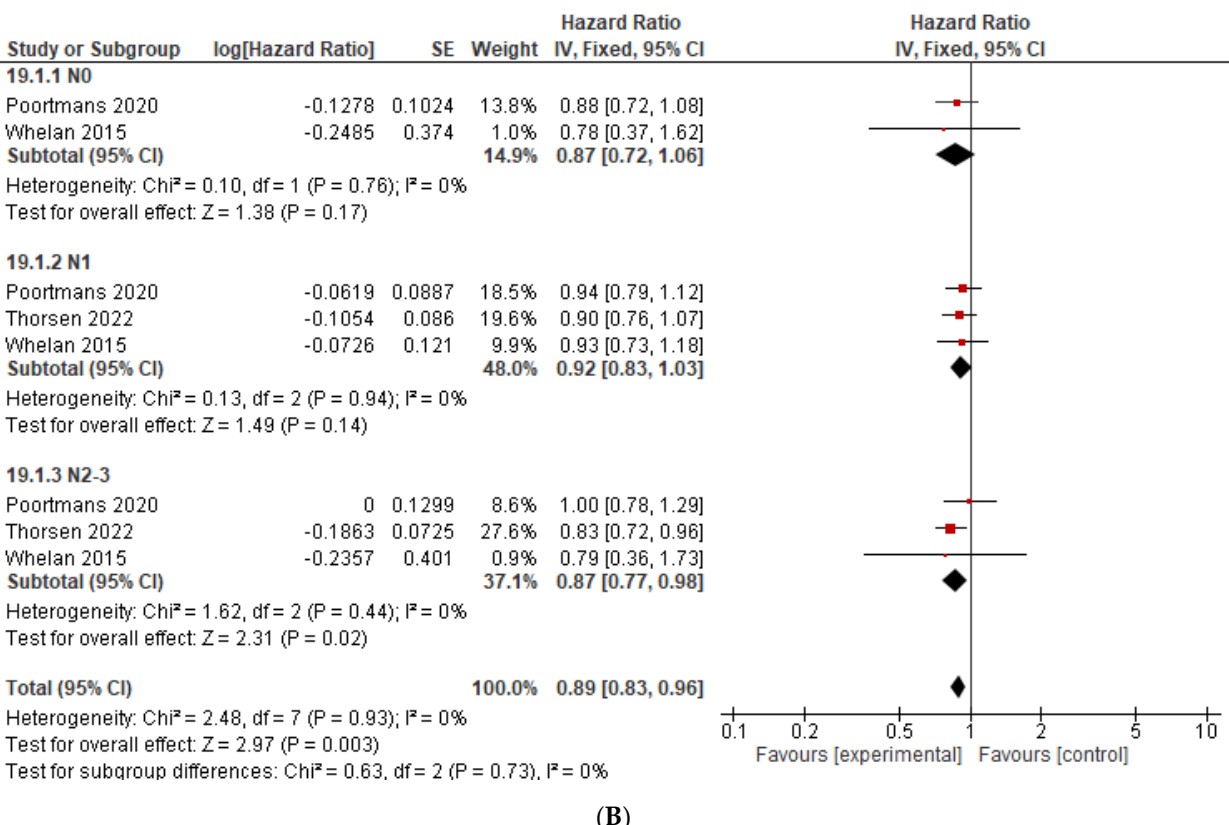

**Figure 4.** Forest plots for OS by subgroups, hazard ratio for: (**A**) OS by tumor location (medial/central compared to lateral); (**B**) OS by nodal involvement. The weight of each trial in the meta-analysis is represented by the size of the squares. The estimated pooled effect is represented by a diamond shape. *p* values are two-sided. References: Hennequin 2013 [13], Kim et al., 2021 [24], Poortmans 2020 [12], Thorsen 2022 [15], Whelan 2015 [11].

The odds for second malignancy were similar between the IMNI and the control group (OR = 0.95, 95% CI 0.82–1.10, *p* = 0.51), see Figure 5A. IMNI was associated with increased odds of cardiovascular morbidity, which approached, but did not reach significance,

OR = 1.23, 95% CI 0.99–1.55, *p* = 0.07, absolute difference +1.09%, number needed to treat 92, see Figure 5B. There were comparable odds for cardiovascular mortality (OR = 1.00, 95% CI 0.65–1.46, *p* = 1.00) and non-breast cancer related mortality (OR = 1.05, 95% CI 0.79–1.41, *p* = 0.74), see Figure 5C,D. The odds for contralateral breast cancer were comparable between the IMNI and the control group (OR = 1.07, 95% CI 0.77–1.51, *p* = 0.68), see Figure 5E. Data on pneumonitis were not pooled as these were reported only in two studies [11,24]. There were numerically more events of pneumonitis in the IMNI compared to the control group (absolute difference between 1.0 and 2.9%).

Multiple sensitivity analyses for both efficacy and toxicity results have shown similar results, see Table 3. Results of the meta-regression analysis are shown in Table 4. A higher proportion of grade 3 disease, administration of chemotherapy, and less than 10 lymph nodes removed were associated with highly quantitatively significant lower HR for OS (i.e., increased benefit from IMNI). A higher proportion of medial/central tumors, node negative, ER positive, tumor size of 2 cm or smaller, and older women were associated with highly quantitatively significant higher HR for OS (i.e., lower benefit from IMNI).

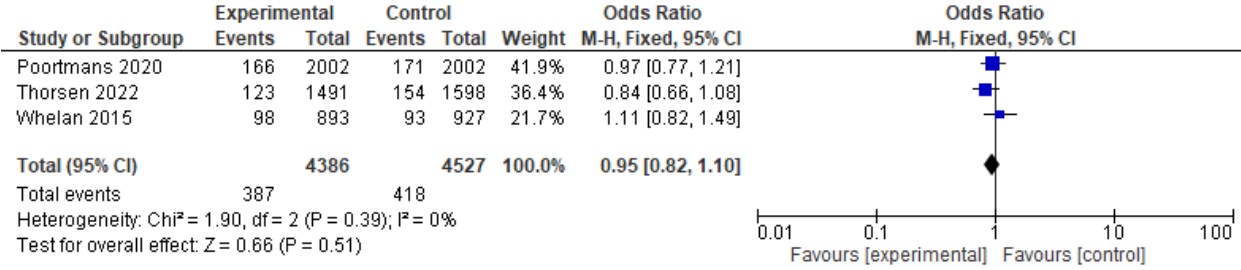

(A)

(B)

(C)

**Figure 5.** *Cont.*

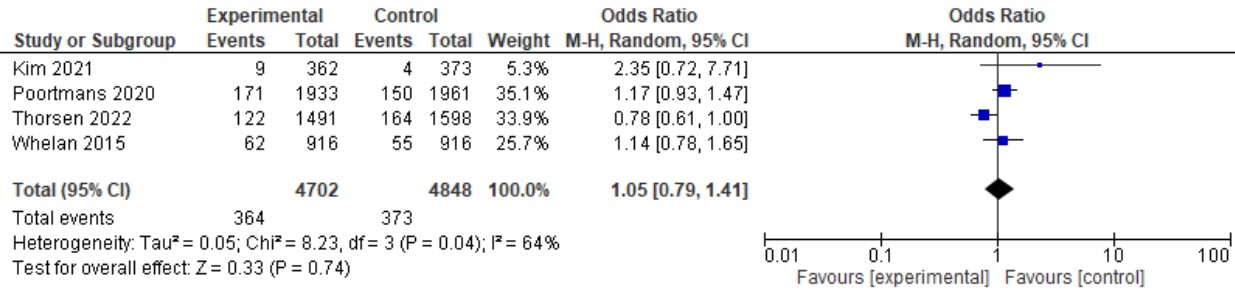

**(D)**

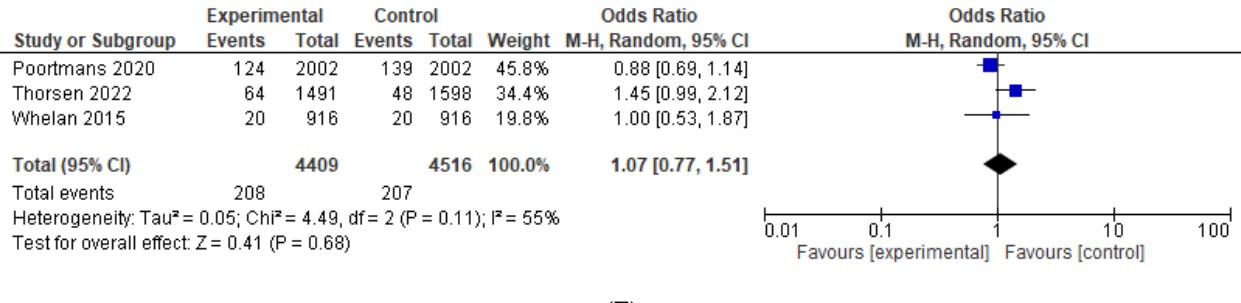

**(E)**

**Figure 5.** Forest plots for the toxicity, odds ratio for: (**A**) second malignancy; (**B**) cardiovascular morbidity; (**C**) cardiovascular mortality; (**D**) non-breast cancer related mortality; (**E**) contralateral breast cancer. The weight of each trial in the meta-analysis is represented by the size of the squares. The estimated pooled effect is represented by a diamond shape. *p* values are two-sided. References: Hennequin 2013 [13], Kim et al., 2021 [24], Poortmans 2020 [12], Thorsen 2022 [15], Whelan 2015 [11].

**Table 4.** The results of the meta-regression for overall-survival.

| Overall-Survival—HR | β | *p* |
|:---:|:---:|:---:|
| Median follow-up | +0.207 | 0.738 |
| Age | +0.906 | 0.278 |
| % Left sided tumors | −0.269 | 0.827 |
| % Medial/central tumors | +0.837 | 0.163 |
| % T ≤2 cm | +0.711 | 0.289 |
| % N negative | +0.952 | 0.048 |
| % Grade 3 | −0.836 | 0.37 |
| % ER positive | +0.63 | 0.937 |
| % Chemotherapy treatment | −0.937 | 0.063 |
| % Mastectomy | −0.04 | 0.96 |
| % <10 lymph nodes removed | −0.777 | 0.434 |

Node negative disease was associated with a significantly greater relative magnitude of effect on OS (*p* = 0.048). Chemotherapy administration had a lower magnitude of effect on OS, where this finding approached, but did not meet statistical significance (*p* = 0.063). All of the other evaluated variables including age, tumor size, grade, extent of surgery, proportion of hormone receptor positive disease and of left sided tumors had no significant impact on OS, BCSS, and non-breast cancer mortality.

## 4. Discussion

The inclusion of IMNI as an integral part of adjuvant radiation planning has for years been under debate. Individual studies have not shown a clear OS benefit and with increased toxicity, the balance between the benefits and risks has remained uncertain. This meta-analysis, comprising the data of five prospective controlled trials studies with a weighted median follow of more than 13 years demonstrated that IMNI was associated with a small magnitude, but statistically significant improvement in OS as well as significantly improved BCSS, DMFS, and locoregional recurrence together with comparable non-breast cancer mortality.

Despite a statistically significant improvement in OS, the magnitude of improvement was modest, with only a 9% relative improvement associated with IMNI. This magnitude of effect is of questionable clinical meaningfulness. It should be noted that the population included in this meta-analysis is heterogenous, also comprising patients with relatively low risk for IMN involvement. This included women with node negative disease or women with lateral tumors together with a low burden of nodal involvement. Current guidelines consider adding IMNI in selected patients including medial/central situated tumors or disease with a high burden of nodal involvement (i.e., pN2-3) [8,9,26]. These guidelines are based on results from individual trials, suggesting higher IMNI benefit in these specific subgroups [4,5,14]. While subgroup analyses in this current meta-analysis failed to identify specific subgroup with greater relative benefit, the meta-regression results did identify a potential higher magnitude of benefit from IMN in higher risk populations including a higher proportion of patients with grade 3 disease, node positive disease, larger tumors, and less patients with ER positive disease as well as in younger women. Meta-regression results also supported increased benefit in the relative benefit from IMNI in studies with a higher proportion of chemotherapy administration and in studies with less extensive nodal removal. Overall, in patients with higher absolute risks of recurrence, it is expected to achieve a clinically meaningful benefit. Of note, studies including only node positive disease had a higher benefit from IMNI compared to studies that also included patients without nodal involvement (HR for OS 0.86 compared to 0.94), though this difference was not statistically significant. Medial/central tumors are associated with increased risk for IMN metastasis [5], however in this meta-analysis, the medial/central and lateral tumors showed a comparable OS benefit with IMNI compared to lateral tumors in subgroup analysis. Interestingly, meta-regression results showed that a higher proportion of medial/central tumors was associated with a lower magnitude of benefit from IMNI.

Adjuvant irradiation has a well-established role in omitting axillary lymph node dissection in patients with a low burden of clinical involvement who presented with clinical node negative disease [27,28] and the ongoing Alliance 011202 trial is aiming to answer whether RNI including IMNI is non-inferior to axillary lymph node dissection in patients who presented with node positive disease [29]. We found a trend for higher relative benefit from IMNI in patients with less extensive axillary lymph node removal and this signal was also demonstrated by the meta-regression results, supporting the role of RNI in patients whose axillary surgery was deescalated. Results from the randomized prospective studies are awaited in order to adopt de-escalated surgery to the axilla routinely.

There was heterogeneity between the studies included in this meta-analysis with respect to the treatment that was given to the control groups, the type of surgery to the breast, and the population included in individual studies. Additionally, one study was a prospective non-randomized study [14]. To address this heterogeneity, we conducted multiple sensitivity analyses that showed similar results, suggesting that overall conclusions from the main analysis are likely to be robust. While irradiation to supraclavicular fossa with/without irradiation to the axilla is considered as the standard of care for high-risk disease [26]; in two studies, the control group was treated with potentially inferior treatment as radiation was given only to the breast/chest wall fields [10,11]. However, subgroup analysis by the extent of radiation in the control group demonstrated a comparable OS benefit from IMNI.

The main concern when adding IMNI is increasing the long-term toxicity. Cardiac toxicity, pneumonitis, and secondary malignancies are all associated with adjuvant breast irradiation [30–32], and the odds for these adverse events are increased with higher doses and more extensive radiation fields [32–35]. Our analysis demonstrated a relative excess of cardiovascular morbidity of 23%; this finding was not statistically significant, however, power to identify the significance of individual toxicity was low and this relative increase translated to a >1% absolute risk of cardiovascular morbidity albeit without translating to cardiovascular mortality. As long-term toxicity associated with adjuvant radiation may present many years after treatment [33], accurate characterization of long-term cardiovascular morbidity may be challenging. Of note, improvement in radiation technology over time has led to increased efficacy and decreased toxicity in early-stage breast disease. Results from the Early Breast Cancer Trialists' Collaborative Group (EBCTCG) meta-analysis have shown that over the years, there was a favorable change in the risk–benefit balance from adjuvant RNI in early-breast cancer [36]. This can be attributed to the decreasing mean heart dose in trials performed in later years [36]. Today, all radiation treatment plans are computer tomography (CT)-based and utilize the delineation of organs at risk, both strategies have markedly improved the target coverage and sparing heart and lung dose [37–39]. Moreover, breath-hold strategies, which are considered as the standard of care in patients receiving radiation treatment to the left breast, have enabled the receipt of radiation with significantly lower heart and lung dose, taking advantage of the anatomical change of lung expansion, which was demonstrated in patients receiving radiation with and without IMNI [40–42]. Implementing advanced modalities such as intensity-modulated radiotherapy with breath holding or intensity-modulated proton therapy with free breathing may also significantly lower heart dose [43]. These changes in radiation planning will hopefully translate to less cardiovascular morbidity in the future, positively influencing the risk–benefit balance of IMNI.

This meta-analysis had several limitations. First, this was a literature-based rather than an individual patient meta-analysis. Second, as breast/chest wall radiation only was the treatment of the control group in two studies included in this meta-analysis [10,11], the interpretation of the effect of the addition of IMNI to other RNI fields is more prone to uncertainty. Third, the included studies were heterogenous with regard to the included population and the extent of the surgery. However, sensitivity analyses addressing this limitation did not identify the effect-modifying variables. Fourth, data on the baseline comorbidities and surgical techniques were not available and could not be adjusted. Additionally, adequate monitoring for toxicity over the duration of follow-up was uncertain, and therefore, it is possible that toxicity was underestimated. With modest survival improvement with IMNI, the risk–benefit balance could be influenced by longer term (and unreported) toxicity.

## 5. Conclusions

In high-risk early breast cancer, IMNI was associated with statistically significant improvement in OS, BCSS, DFS, and DMFS compared to the control group. Considering the modest benefit and the potentially increased toxicity, it is unclear whether IMNI has clinically meaningful benefit in all patients. Subgroup analysis in this meta-analysis did not identify the sub-population that clearly has a higher magnitude of benefit, though the meta-regression results did suggest a higher relative benefit for women with unfavorable clinic-pathological characteristics and an overall higher risk for recurrence are expected to drive higher absolute benefit from IMNI. With the improvements in the technology of breast irradiation likely resulting in lesser toxicity, the addition of IMNI in selected high-risk populations seems reasonable. However, individual decision making is desirable for each patient considering the clinical risk, age, comorbidities, and meticulous radiation planning.

**Supplementary Materials:** The following supporting information can be downloaded at: https://www.mdpi.com/article/10.3390/curroncol29090523/s1. Figure S1—Forest plots for overall survival by study design, hazard ratio for: A: Overall survival by inclusion of node negative disease- inclusion of only node positive disease compared to inclusion of both node negative and node positive disease. B: Overall survival by extend of treatment to the control group- radiation to the control group comprised breast/ chest wall only compared to breast/ chest wall and other regional lymph nodes.

**Author Contributions:** Conceptualization, Y.K. and H.G.; Methodology, Y.K., E.A. and H.G.; Validation, Y.K., D.L. and H.G.; Formal Analysis, H.G.; Investigation, Y.K., D.L., A.G., B.W.C., E.A. and H.G.; Data Curation, Y.K., D.L. and H.G.; Writing—Original Draft Preparation, Y.K. and H.G.; Writing—Review & Editing, Y.K., D.L., A.G., B.W.C., E.A. and H.G.; Visualization, H.G.; Supervision, H.G. All authors have read and agreed to the published version of the manuscript.

**Funding:** The authors received no financial support for the research, authorship, and/or publication of this article.

**Institutional Review Board Statement:** Ethical review and approval were waived for this study as this was a systemic review and meta-analysis using the existing published data.

**Informed Consent Statement:** Patient consent was waived as this was a systemic review and meta-analysis using existing published data.

**Data Availability Statement:** The data presented in this study are available on request from the corresponding author.

**Conflicts of Interest:** Benjamin W. Corn is the Chief Medical Officer of Lutris Pharma. Eitan Amir reports consulting fees from: Exact Sciences, Sandoz, and Novartis, all outside the submitted work. Hadar Goldvaser reports personal fees from: Gilead (honorarium and consulting), Eli-Lilly (honorarium and consulting), MSD (honorarium), Novartis (honorarium and consulting), Pfizer (honorarium), Rhenium Oncotest (honorarium), and Roche (honorarium), all outside the submitted work. All other authors have no conflicts of interest.

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
