# Peer review of "Efficacy and Safety of the Addition of Internal Mammary Irradiation to Standard Adjuvant Radiation in Early-Stage Breast Cancer: A Systematic Review and Meta-Analysis"

_curroncol, doi:10.3390/curroncol29090523_

Round 1
Reviewer 1 Report
Adjuvant radiation therapy (RT) is an integral part in the management of breast cancer. In patients with node positive disease or high risk node negative disease, breast/chest wall and nodal irradiation is commonly performed. Nodal irradiation is often limited to the supraclavicular nodes and the axillary region at risk. The addition of internal mammary nodes (IMN) irradiation is still controversial, as you have highlighted.
Several studies have been published on this topic, but only some of them are clinical trials. Moreover, systematic reviews and meta-analyses are rare. I have mainly appreciated your study for this reason.
The title is appropriate and you have accurately reported on the recent literature and on the clinical trials you have analysed. The text is clear and exhaustive and you have critically discussed the results of the meta-analysis.
On the whole I think it is a very interesting analysis: the comparison of randomized trials offers an intriguing option to investigate previously not directly compared treatment options.
In conclusion this study is of relevance and interest to the scientific community and I don’t suggest any adjustment to the manuscript.
Author Response
I thank you for your manuscript evaluation and review.
Reviewer 2 Report
Reviewer's report
Manuscript ID: CO-1878631
Title: Efficacy and Safety of the Addition of Internal Mammary Irradiation to Standard Adjuvant Radiation in Early-Stage Breast Cancer: A Systematic Review and Meta-analysis
Date:2022/9/5
Reviewer's report:
This is an interesting manuscript as its a comprehensive study of the benefit of IMNI on early stage breast cancer. Current standard of care supports regional nodal irradiation (RNI) in most node positive breast cancer patients, in addition to whole breast or chest wall irradiation. At present, patients with medially or centrally located tumors may benefit from the use of IMNI, other than that, including IMNI as a standard regional LN irradiation remain controversial as long term side effect outweight the treatment beneift. As this study have also proven a modest increase OSR associated with IMNI. And the high rate of cardiotoxicity (23%) compare with non-IMNI group.
The MS is well prepared and containing a large amount of data. Although, there remain several limitation such as it was a literature base studies, heterogenous study population, Toxicity of IMNI was not addressed. Surgical technique was never explained either. Since there was wide differences in OSR, disease-free survival rate for difference axilary LN surgical technique. Nevertheless, it was still well written, thus, could be consider for publication.
Author Response
- We agree with the reviewer that our meta-analysis has several limitations as detailed above. These limitations are specified in the discussion section, see page 21, second paragraph. Of note, several sensitivity analyses aiming the address this heterogeneity showed similar results to the primary analysis results.
“This meta-analysis has several limitations. First, this is a literature-based rather than an individual patient meta-analysis…... Third, included studies were heterogenous with regards to the included population and the extent of surgery. However, sensitivity analyses addressing this limitation did not identify effect-modifying variables.”
- With regards to toxicity: Data on meaningful adverse events that have a potential to affect long term outcome were extracted, including cardiotoxicity and cardiovascular mortality, secondary cancer and pneumonitis. Data on pneumonitis were reported only in 2 studies, and therefore were not pooled. Other toxicity related results are presented in pages 16-17.
- We knowledge specific surgical technique were not reported. However, we did report data on d rates of breast conserving surgery and mastectomy and the on the extend of the surgery to axilla. These results are shown in Table 2. The limitation regarding missing data on surgical technique were elaborated to the discussion section, see page 21, second paragraph:
“Fourth, data on baseline comorbidities and surgical techniques were not available and could not be adjusted.”
